# Emergence of the Synucleins

**DOI:** 10.3390/biology12081053

**Published:** 2023-07-27

**Authors:** Ignacio Marín

**Affiliations:** Instituto de Biomedicina de Valencia, Consejo Superior de Investigaciones Científicas (IBV-CSIC), 46010 Valencia, Spain; imarin@ibv.csic.es; Tel.: +34-963-393-770

**Keywords:** synuclein, whole-genome duplication, agnathans, gnathostomes, Parkinson’s disease

## Abstract

**Simple Summary:**

Alpha-synuclein has been thoroughly analyzed due to its relevance to familial Parkinson’s disease and other synucleinopathies. In this study, I determine the origin of the synuclein genes in all vertebrates. Contrary to previous assumptions, these genes are not the result of individual gene duplications. They are ohnologs that emerged in several whole-genome duplications that occurred throughout vertebrate history.

**Abstract:**

This study establishes the origin and evolutionary history of the synuclein genes. A combination of phylogenetic analyses of the synucleins from twenty-two model species, characterization of local synteny similarities among humans, sharks and lampreys, and statistical comparisons among lamprey and human chromosomes, provides conclusive evidence for the current diversity of synuclein genes arising from the whole-genome duplications (WGDs) that occurred in vertebrates. An ancestral synuclein gene was duplicated in a first WGD, predating the diversification of all living vertebrates. The two resulting genes are still present in agnathan vertebrates. The second WGD, specific to the gnathostome lineage, led to the emergence of the three classical synuclein genes, *SNCA*, *SNCB* and *SNCG*, which are present in all jawed vertebrate lineages. Additional WGDs have added new genes in both agnathans and gnathostomes, while some gene losses have occurred in particular species. The emergence of synucleins through WGDs prevented these genes from experiencing dosage effects, thus avoiding the potential detrimental effects associated with individual duplications of genes that encode proteins prone to aggregation. Additional insights into the structural and functional features of synucleins are gained through the analysis of the highly divergent synuclein proteins present in chondrichthyans and agnathans.

## 1. Introduction

Between 1988 and 1998, a group of vertebrate-specific proteins, now known as synucleins, was discovered in mammals, birds and fishes [1,2,3]. In gnathostome vertebrates, including humans [4], three of these proteins, named alpha-, beta- and gamma-synuclein (hereafter abbreviated as α-syn, β-syn and γ-syn, respectively) exist. Additional synuclein genes have been detected in a few species, all of which are recent duplicates [5,6,7,8]. Synucleins are short proteins, typically 120–145 amino acids long, with a peculiar structure: imperfect 11-amino-acid-long repeats are observed at the N-terminus, while the 30–40 amino acids closest to the C-terminus lack any repetitious sequences and include clusters of acidic residues [4,9,10]. Synucleins remain unstructured in solution, but their N-terminal repetitious sequences form amphipathic α-helical structures when coupled with lipids [11,12]. For α-syn, it has been shown that, after such binding and depending on the particular conditions, either a single, long α-helix or two shorter α-helices, separated by a small link, are formed [13,14,15,16,17]. The α-helices in synucleins are structurally similar to those detected in apolipoproteins [18,19]; both the apolipoprotein and the synuclein helical regions insert into membranes to induce their curvature [20,21]. The cellular functions of the synucleins are multiple and still incompletely understood. Alpha-synuclein is highly expressed in the adult brain of all gnathostomes analyzed, i.e., mammals, birds, reptiles, amphibians and fishes [5,22,23,24,25,26,27,28]. Expression greatly varies in different regions; in a comprehensive study of the rat brain, it was concluded that α-syn levels were high in the catecholaminergic system and low in cholinergic regions [29]. The α-syn protein is enriched in presynaptic terminals, where it has several roles in regulating synaptic transmission, such as the modulation of neurotransmitter release, the organization of vesicle pools and vesicle trafficking, as well as acting as chaperone of other proteins involved in synaptic function [30]. There is evidence for α-syn also being involved in regulating mitochondrial function, protecting the cells against oxidative stress and apoptosis and in lipid transport [31]. Much less is known about the other synucleins. Beta-synuclein is also highly expressed in the brain of all the gnathostomes analyzed [5,6,10,25,27,32,33]. In rodents, it has been shown that its levels of expression in different regions of the brain are, on average, higher and also more homogeneous than those of α-syn [29,34]. In rat brain, although α-syn and β-syn are found together in many regions, high β-syn expression is found in cholinergic areas where α-syn expression is low [29]. The presence of both proteins in some cells may have important functional consequences, given the evidence for β-syn acting as an antagonist/regulator of α-syn function [35,36]. Other roles for β-syn, such as regulating apoptosis and the pathways of protein degradation or an involvement in the control of metal levels, have been suggested [36]. Gamma-synuclein has a pattern of expression that is quite different from the other two members of the family. Although it is also expressed in the brain, it is predominantly found in the peripheral nervous system and is also detected at relatively high levels in some non-neural tissues [4,5,6,27,28,29,37,38]. In fact, γ-syn was initially discovered because of its high expression in breast cancer cells [39]. Since then, it has been associated with the progression of several types of cancer [40,41].

Some dominant, missense mutations, as well as duplications or triplications of the human *SNCA* gene, which encodes α-syn, cause familial Parkinson’s disease (PD) [42,43,44,45]. The abnormal folding of large quantities of α-syn proteins, resulting in intracellular toxic fibrils and aggregates, explains its involvement in PD and other neurodegenerative diseases, such as dementia with Lewy bodies (DLB) or multiple system atrophy (MSA), collectively known as synucleinopathies [46,47]. In fact, Lewy bodies, the characteristic protein aggregates found in PD and other synucleinopathies, contain large amounts of α-syn fibers [48]. A central, hydrophobic region of α-syn (amino acids 61–95 in the human protein, which includes its two most C-terminal 11-amino-acid-long repeats), known as NAC [23], plays a critical role in aggregation [49,50,51]. Misfolded α-syn proteins can spread between cells, contributing to the progression of synucleinopathies in a prion-like manner [52,53]. The involvement of the immune system and neuroinflammation in these diseases is also well established [54]. Interestingly, mutations in *SNCB*, the gene that encodes β-syn, may also contribute to DLB [55,56].

In humans, the synuclein-coding genes are located on three different chromosomes; *SNCA*, which encodes α-syn, is located at region 4q.22, *SNCB* (β-syn) at 5q.35, and *SNCG* (γ-syn) at 10q.23 [4,57]. Similar patterns have been observed in other species; the synuclein genes have never been found clustered together. The three genes have complex, similar exon/intron structures [2], indicating that they did not originate from retroposition events. The sequence and exon/intron structure similarities among the three *SNC* genes indicate that they emerged from a single ancestor gene that already had features similar to those found in the genes of current species. Because α-syn and β-syn are more similar to each other than either is to γ-syn [2], it has been proposed that the three genes originated from two conventional duplications of a single progenitor gene, the first generating an *SNCG* gene and an *SNCA/B* gene, and a second duplication, affecting the *SNCA/B* gene, which produced the modern *SNCA* and *SNCB* genes [7,58]. The dispersion of these three genes in vertebrates would be due to gene transpositions [7,58]. The exact timing of these putative duplications has not been established, but there is no reason to think that both occurred before the last common ancestor of all living vertebrates was born, because several studies suggested that lampreys may have only γ-syn encoding genes [7,59,60]. Therefore, the hypothesized *SNCA/B* progenitor could potentially still be found in particular gnathostome lineages, a fact that, if true, would strongly support the accepted model of synuclein evolution. Unfortunately, the available information about how synucleins evolved is still too fragmentary to establish whether *SNCA/B* genes exist or not. For example, it is not known whether α-, β- and γ-synucleins are present in all vertebrate lineages. Previous studies provided evidence for the existence of the three synucleins in several placental mammals, birds and teleostean fish species [2,3,5,7,8,61], species of two reptile genera (*Anolis*, *Chelonia*; [7]), amphibians of the *Xenopus* genus [5,7] and a single chondrichthyan, the Australian ghostshark, *Callorhinchus milii* [8,60]. However, all these studies have significant limitations. First, it is possible that additional synucleins were missing in some species due to incomplete genomic data. Some studies detected only one or two synuclein genes in certain species (see, e.g., [7]), but it is unclear whether the missing genes were truly absent or just unknown at that time. Second, the methodologies used to obtain multiple-sequence alignments and phylogenetic trees have greatly improved since those studies were published. This is critical, because suboptimal approaches can lead to artifactual tree topologies, given that synucleins are small proteins, i.e., the amount of useful information provided by their sequences is limited. Also, the C-terminal end of synucleins evolves rapidly, making it challenging to obtain reliable alignments for that region. It is also important to note that only two of the published studies [7,61] provided information on the statistical support for the tree topologies, which is essential for determining the accuracy of the proposed orthologies. Lastly, none of these studies incorporated together sequences from all the main vertebrate taxa. These methodological issues mean that, while there is no doubt that the currently accepted orthologies are correct for closely related species, they may be erroneous for distant relatives.

In this work, I analyze the evolution of the synucleins in all major vertebrate groups, using 22 model species, and propose a new hypothesis regarding the origin of synuclein genes. Briefly, I suggest that synuclein genes are ohnologs that emerged in the whole-genome duplications (WGDs) that occurred in early vertebrate evolution (Figure 1).

This hypothesis depends on recent studies that have established the occurrence of the first of those WGDs (WGD1) before the agnathan/gnathostome split, and the second (WGD2) just after the split, preceding gnathostome diversification [62,63]. Furthermore, evidence has been found for additional WGDs having occurred early in agnathan evolution, prior to the divergence of lampreys and hagfishes [63,64,65], which may have contributed to an increase in the number of synuclein genes. This new hypothesis generates three major predictions (also summarized in Figure 1). As we will see, evidence supporting the validity of all three predictions refutes the current model of two successive, conventional duplications to explain the emergence of synuclein genes.

## 2. Materials and Methods

***Sequence retrieval and multiple-sequence alignments:*** Twenty-two species representing all the major vertebrate lineages were selected for the subsequent analyses (Table 1).

This set includes species from taxa that have not been hitherto explored, such as non-eutherian mammals, crocodiles, sturgeons, bichirs, and lungfishes. Additionally, a representative from each of the three main chondrichthyan clades (chimaeras, sharks and rays) was chosen. For agnathans, five species were analyzed. The genome of the sea lamprey *Petromyzon marinus* (assembly kPetMar1.pri; [66]) can be considered nearly fully sequenced and assembled. The genomes of three other agnathans, namely the lampreys *Lethenteron camtschaticum* and *Entosphenus tridentatus*, and the hagfish *Eptatretus burgeri*, which have been deeply but not completely sequenced [63,64,65,67,68], were also included in the analysis. Furthermore, the synuclein genes characterized in a fourth lamprey, *Lampetra fluviatilis* [60] were also incorporated. It is possible that a few relevant sequences from the genomes of these four incompletely analyzed species may still be missing. Therefore, it is advisable to compare them with the sequences found in *Petromyzon* to gain a better understanding of the diversity of synucleins in the agnathan lineage.

The synuclein protein sequences present in those 22 species were retrieved from the nucleotide collection (nr/nt), whole-genome shotgun contigs (wgs), transcriptome shotgun assembly (tsa) and expressed sequence tags (est) databases of the National Center for Biotechnology Information (NCBI) using the protein sequences of the three human synucleins as queries in TBLASTN searches. Additional searches performed with synucleins from several gnathostome and agnathan species determined that those human-specific searches had already detected all the available synuclein sequences in the 22 model species. When almost identical sequences were found that corresponded to variations of the same gene in different individuals of a given species, the ones compiled in the nr/nt database, when present, were chosen. Otherwise, the ones in the tsa database were preferred. The N-terminal region of all synucleins is very similar and relatively easy to align. However, the acidic C-terminal tails of α- and β-synucleins are quite similar, but significantly different from the tails found in γ-synucleins. Given the difficulty of correctly aligning these regions, alternative multiple-sequence alignments were generated using the programs ClustalX 2.1 [69] and MAFFT 7.505 [70]. For ClustalX, an alignment with its default parameters (pairwise alignment parameters: Gonnet 250 matrix; gap opening penalty = 10; gap extension penalty = 0.1. Multiple alignment parameters: Gonnet matrix series; gap opening penalty = 10; gap extension penalty = 0.2; delay divergent sequences = 30%) was obtained. For MAFFT, the six available alignment strategies in the program (FF-TNS-1, FF-TNS-2, FFT-NS-i, L-INS-i, E-INS-i and G-INS-i) were used. In the FFT-NS-i, L-INS-i, E-INS-i and G-INS-i analyses, the number of cycles of iterative refinement was set to 1000.

***Phylogenetic analyses and selection of the optimal trees:*** Maximum-likelihood (ML) phylogenetic analyses were performed using IQTREE 2.2.0 [71]. As shown in several works, it is essential to use different perturbation strengths (i.e., several values of the *pers* parameter used by IQTREE) and to repeat the tree search multiple times (*runs* parameter > 1) to obtain an optimal ML tree. In accordance with the recommendations of the developers of the program [71,72] and as already implemented in my previous works [73,74], the following parameters were used: *pers* was alternatively set at 0.2, 0.5 or 0.8; number of replicates to stop the analysis (*nstop* parameter) equal to 500; the number of independent runs (*runs* parameter) was set at 10. The best protein evolution model was determined using ModelFinder [75]. The number of ultrafast bootstrap replicates (*bb* parameter; [76]) to determine the reliability of the topology obtained was set at 1000. ML trees were generated for each of the seven alternative ClustalX or MAFFT alignments. Thus, a total of 21 alternative ML trees (i.e., 7 different alignments × 3 perturbation strengths) were obtained for each dataset. The tree with the highest likelihood value was considered optimal; given that the same model of protein evolution was selected as best by ModelFinder for all ML analyses and that all the alignments had very similar lengths, using other criteria to select the optimal tree, such as the Bayesian information criterion [75], led to the same conclusions. Subsequently, a truncated version of the optimal alignments, which included only the highly conserved N-terminus (corresponding to amino acids 1–97 in human α-syn) was used for additional IQTREE analyses with the same *pers*, *nstop*, *runs* and *bb* parameters. This was done in order to establish whether including the highly variable C-terminal end of synucleins could lead to the recovery of spurious relationships among the sequences, due to a wrong alignment of that end or convergent evolution of the C-terminal sequences of very different synucleins. Tree figures were drawn with Mega 7.0.26 [77].

***Synteny analyses:*** Reconstructing the regions where the synuclein genes are located in different species can contribute to supporting or refuting whether these genes are orthologs, given that it is expected that orthologous genes will generally be found on the same chromosomes and often surrounded by the same genes, even in distantly related organisms. In this study, the genomic regions containing synuclein genes were reconstructed for three species: *Homo sapiens*, the shark *Rhincodon typus* and the lamprey *Petromyzon marinus*. These species were selected as representatives of osteichthyans, chondrichthyans and agnathans, respectively. The five protein-coding genes located on each side of the three synuclein genes in each species were obtained from the NCBI gene database. To determine the most likely human orthologs, the fish genes were compared with the set of proteins encoded by human genes included in the reference protein database (Refseq_protein) using BLASTP searches.

***Chromosomal comparisons:*** Two analyses were conducted involving the comparison of protein-coding genes across whole chromosomes. The first analysis aimed to establish the number of related ohnologs on specific human chromosomes. To achieve this, a list of predicted human ohnologs was downloaded from the OHNOLOGS database (http://ohnologs.curie.fr/ (accessed on 26 July 2023; intermediate criteria list [78]). Additionally, a second list containing all the human protein-coding genes was obtained from the NCBI (https://ftp.ncbi.nlm.nih.gov/refseq/H_sapiens/annotation/GRCh38_latest/refseq_identifiers/GRCh38_latest_protein.faa.gz; accessed on 26 July 2023). By comparing these two lists, all the ohnologs present on chromosomes 2, 4, 5, 8 and 10 were identified. The second general analyses involved comparing all the protein-coding genes located on specific chromosomes of *Petromyzon marinus* with the complete set of human genes. For *P. marinus*, the proteins encoded by all the genes located on three chromosomes where synuclein genes are found (numbers 8, 22 and 41) were downloaded from the corresponding NCBI website (https://www.ncbi.nlm.nih.gov/genome/?term=Petromyzon; accessed on 26 July 2023). These proteins were then compared with the complete set of human proteins, again performing BLASTP searches against the human sequences compiled in the Refseq_protein database.

## 3. Results

In the following sections, evidence will be presented to support the validity of the three predictions derived from the model depicted in Figure 1.

### 3.1. Prediction 1: The Ancestor of All Living Gnathostomes Already Possessed SNCA, SNCB and SNCG Genes; Some Recent, Lineage-Specific Duplications or Losses May Have Occurred

In the model of successive, independent duplications, there is no requirement for all gnathostomes to possess the three synuclein genes. For example, if, as suggested in previous works [7,8,59,60], lampreys only have *SNCG* genes, it is conceivable that some early-branching gnathostomes (e.g., chondrichthyans) have either *SNCG* alone or *SNCG* and its initial duplicate, the *SNCA/B* gene from which *SNCA* and *SNCB* later originated. In contrast, the model depicted in Figure 1, which provides a precise timeline for the emergence of these genes, necessitates the presence of the three synuclein genes in all gnathostome lineages. If this is not observed, the model would be immediately refuted.

Although some studies have suggested the presence of the three synucleins in chondrichthyans [8,60], the evidence provided was weak. The trees generated lacked statistical support and exhibited topologies that did not fit with the known evolutionary relationships among the analyzed species. The first conclusive evidence supporting the presence of the three synucleins in chondrichthyans is presented in Figure 2 and Figure 3: Figure 2 (next page) summarizes the phylogenetic trees obtained when analyzing either the conserved repeat region Figure 2a or the complete synuclein sequences Figure 2b. The corresponding alignment can be found in Appendix A. These trees explain why previous studies did not yield clear-cut results. They suggest that the three synucleins found in the selected chondrichthyan species are the same present in osteichthyans, but evidence is not conclusive. In the tree depicted in Figure 2a, full support (bootstrap = 100%) is obtained for a branch that groups together osteichthyan γ-synucleins and one type of chondrichthyan sequences. However, although the other two chondrichthyan synucleins appear together with osteichthyan α- and β-synucleins, it is not possible to determine their precise orthology relationships, due to low statistical support for all the critical branches. The tree shown in Figure 2b, which incorporates more information by including the N-terminal end of these proteins, strongly suggests the presence of the three synuclein genes in chondrichthyans. However, although now the topology found is exactly the one expected if this is true, the branches including osteichthyan α-syn and β-syn and their most similar chondrichthyan sequences still do not have significant (≥95%) bootstrap support. Considering these results, which may indicate that a formal characterization of the orthology relationships among osteichthyan and chondrichthyan synucleins is impossible using sequence data alone, it is clear that additional, complementary information is required. Figure 3 presents synteny results that conclusively demonstrate the existence of bona fide α-, β- and γ-synucleins in chondrichthyans. By analyzing which human genes exhibit the highest similarity to those surrounding the synuclein-coding genes in chondrichthyans (in this example, the whale shark, *R. tipus*), a striking congruence in the order and polarity of these genes in both humans and sharks can be demonstrated. The combined probability of obtaining the results shown in Figure 2b and Figure 3, which are entirely independent (the first being based on sequence similarity and the second on gene locations), if the genes of chondrichthyans and osteichthyans are not orthologous is obviously negligible.

When considering the entire set of gnathostome sequences in light of the results shown in Figure 2 and Figure 3, it becomes apparent that several species have experienced losses or duplications of synuclein genes. Specifically, three independent losses of the SNCA gene were identified among the 17 analyzed model gnathostomes, in the monotreme mammal Ornithorhynchus anatinus, the holostean fish Lepisosteus aculeatus and the teleostean fish Danio rerio. Given that these three genomes have been deeply analyzed, it is highly unlikely that these genes are still present. No losses were detected for either the SNCB or SNCG genes in our model gnathostomes. Duplications were observed only for the SNCG gene in three actinopterygians: Danio rerio, Takifugu rubripes and Acipenser ruthenus.The duplications in Danio and Takifugu were first detected by Yoshida et al. [5] and are most likely associated with the additional WGD occurred in a common ancestor of both species [79]. Similarly, a recent genome duplication has been detected in the Acipenser lineage [80,81], which may account for the presence of two very similar synuclein genes in A. ruthenus.

### 3.2. Prediction 2: Agnathans Have Two Synuclein Genes, One of Which Is Orthologous to Both the SNCA (α-syn) and the SNCB (β-syn) Gnathostome Genes, While the Other Is Orthologous to Their SNCG (γ-syn) Gene

This is also essential for the model shown in Figure 1 to hold. However, it contradicts the published works suggesting that all cyclostome genes are either *SNCG* orthologs or agnathan-specific genes, lacking a clear relationship with their gnathostome counterparts [7,8,59,60].

Four synuclein genes, from now on referred to as *Agnathan Synuclein-1* to *-4* (abbreviated as *ASYN1-ASYN4*), were identified in agnathan genomes. The phylogenetic trees indicate that these genes can be naturally classified into two distinct classes: *ASYN1* is related to the *SNCA* and *SNCB* gnathostome genes, while *ASYN2-ASYN4* are closely related to gnathostome *SNCG* genes (Figure 4 and Figure 5; see following pages); the corresponding alignments can be found in Appendix A). Agnathan species such as the lampreys *Petromyzon* or *Lethenteron* have all four *ASYN* genes. In the other two lamprey species, one or two genes are missing, but whether these represent true losses or the genes will be discovered once their genomes are fully sequenced is an open question. In the case of the agnathan hagfish, *Eptatretus burgeri*, a full-length sequence was obtained only for *ASYN1* (included in Figure 4 and Figure 5), but short sequences most likely corresponding to two different *ASYN2* genes were also detected. In the genomic sequence of *E. burgeri* with accession number FYBX03000005.1, a region showing 85% identity over 41 amino acids with *Petromyzon ASYN2* was found; additionally, a cDNA (Acc. No. FY413344.1) corresponding to the same gene was also identified (identity = 82% over 61 amino acids when compared with *Petromyzon ASYN2*). In the *Eptatretus* genomic sequence FYBX03000009.1, a second sequence was discovered that may correspond to a slightly more divergent, hagfish-specific *ASYN2* gene (identity = 74% over 42 amino acids when compared with *Petromyzon ASYN2*). These results indicate that hagfishes also possess the two main classes of synuclein genes found in lampreys.

A branch with strong statistical support separates the agnathan *ASYN1* genes and the gnathostome *SNCA* and *SNCB* genes from the other three agnathan genes, *ASYN2-4* and the gnathostome *SNCG* genes (Figure 4 and Figure 5). The simplest explanation for the results summarized in those figures is that agnathans indeed have γ-syn and α/β-syn genes, exactly as predicted in our model. A minor technical point that has to be mentioned is that, as indicated in Figure 1, it would be more appropriate to state that agnathans have, assuming that our model is correct, γ/δ-syn genes (instead of just γ-syn genes). However, since the δ-syn gene, the postulated duplicate of γ-syn emerged in WGD2 (Figure 1), has been lost in all gnathostomes, it is better to omit all references to it in the following discussions.

If Prediction No. 2 of our hypothesis is correct, a corollary is that the chromosome carrying the agnathan α/β-syn gene and those where the gnathostome *SNCA* and *SNCB* genes are located must be evolutionarily related (see Figure 1). Similarly, there should be a correspondence between the chromosomes where γ-syn genes are located in agnathans and gnathostomes (as indicated in the same figure). Given that, if our model is correct, all these chromosomes should derive from a single one that existed before WGD1, it is convenient to draw upon the studies that reconstructed the proto-vertebrate genome and the relationships among the gnathostome and agnathan chromosomes. The published data indicate that the regions containing the synuclein genes may indeed all derive from a single proto-vertebrate chromosome, referred to as CLGQ [62] or Pvc8 [63]. According to those studies, from this ancestral chromosome derived not only fragments of human chromosomes 4, 5 and 10, where our synuclein genes are found, but also portions of the *P. marinus* chromosomes 8, 22 and 41, where the *ASYN1-ASYN3* genes are located. Actually, those authors analyzed the *Petromyzon* scaffolds defined in a previous genome assembly [82], and the correspondence with the current chromosomes is as follows: chromosome 8 = scaf_00012, accession number PIZI01000012.1; chromosome 22 = scaf_00024, Acc. No. PIZI01000024.1; chromosome 41 = scaf_00044, Acc. No. PIZI01000044.1. The fourth gene, *ASYN4*, has been detected in very small contigs in the three lamprey genomes in which it has been found (*P. marinus*, *L. camtschaticum* and *E. tridentatus*) and to which chromosome those contigs correspond has not yet been established.

Thus, these general, previous results are consistent with our model. However, much more detailed predictions can be made that must also hold true for it to be accepted. First, while in the hypothesis of successive independent duplications plus transposition to other chromosomes, the synuclein genes are expected to be surrounded by genes of completely different origins, if their multiplication is linked to the WGDs, as suggested here, then they are all expected to be surrounded by evolutionarily related genes of the same origin. This is shown to be true in Figure 6, in which the ten genes adjacent to the *Petromyzon ASYN1*, *ASYN2* and *ASYN3* genes are detailed. The figure also shows the most likely orthologs in humans, i.e., the top scores in BLASTP searches using the lamprey genes as queries against all human proteins. The results are conclusive: most genes surrounding the lamprey synuclein *ASYN1-ASYN3* genes have likely orthologs located on human chromosomes 4, 5 or 10. For the fourth gene, *ASYN4*, although only the two genes adjacent to it can be determined from the available *Petromyzon* data, the same holds true: their most likely human orthologs are *VCL* (located at 10q22) and *HNRNPH1* (5q35). As shown in Figure 6, all ten genes surrounding *ASYN3* have orthologs in human chromosomes 4, 5 or 10, suggesting that this region may have been more evolutionarily conserved than those around *ASYN1* or *ASYN2*. This is indeed proven by comparing Figure 6 with Figure 3. Orthologs of the human *GRID2*, *CCSER1*, *MMRN1/2*, *GLUD1*, *SHLD2* and *GPRIN1* genes are detected, in similar positions and orientations, around the synuclein genes of humans, sharks and lampreys. Most interestingly, considering all the results shown in those two figures together, it can be deduced that a block of seven genes (*GRID-CCSER-MMRN-SNC-GLUD-SHLD-GPRIN*) was present already in the ancestor of all vertebrates. This block has been strictly conserved on chromosome 8 of *Petromyzon* (Figure 6); about half of it remains around gnathostome *SNCA* and *SNCG* genes (Figure 3), and isolated genes of the block can still be observed close to gnathostome *SNCB* genes (Figure 3) and lamprey *ASYN1* and *ASYN2* genes (Figure 6).

The only way to explain all the results shown in Figure 3, Figure 4, Figure 5 and Figure 6 by successive gene duplications, totally unrelated to both WGD1 and WGD2, is by five independent events all happening. These events would be as follows: (1) A single γ-synuclein gene existed before the gnathostome/agnathan split, located on the ancestral chromosome CLGQ/Pvc8; (2) all agnathan synucleins are γ-synucleins, and three of them emerged after the agnathan/gnathostome split. This would imply that the strong similarity between the *ASYN1* protein and gnathostome α- and β-synucleins (Figure 4 and Figure 5) is due to convergence; (3) The duplications that gave rise to gnathostome α- and β-synucleins, as well as those that generated the three additional agnathan γ-synucleins, involved multiple adjacent genes. Subsequently, several of these duplicate genes transposed together to other chromosomes. This is obviously the only way to explain the presence of related genes around all these synuclein genes on different chromosomes, assuming that the duplications were totally independent; (4) These multiple-gene duplications and transpositions initially involved duplicating segments of the CLGQ/Pvc8 chromosome and, without a single exception, ended up moving the gene blocks to chromosomes that, by chance, also contain genes derived from CLGQ/Pvc8; And, (5) all the synuclein genes that emerged in WGD1 and WGD2 were lost. Of course, all this happening is highly improbable, to say the least. Therefore, the hypothesis of successive duplications unrelated to the vertebrate WGDs can already be considered refuted by the aforementioned data.

The results are in perfect agreement with our model of synuclein evolution (Figure 1). However, an alternative hypothesis may still be proposed. It could be suggested that gnathostome synucleins indeed emerged as a result of WGD1 and WGD2, but agnathans only have γ-synucleins, because the ancestral α/β-syn gene that emerged after WGD1 was lost in the agnathan lineage, while additional γ-syn genes emerged, later and independently, in agnathans. The presence of multiple blocks of similar genes surrounding the gnathostome genes would be explained by their emergence in WGD1 and WGD2. However, the presence of similar genes around the agnathan synuclein-coding ones (Figure 6) would have a different explanation: the agnathan-specific genome hexaploidization that occurred after the gnathostome/agnathan split [63]. This alternative hypothesis could also account for all the results presented in this study thus far, given that, contrary to what happened for humans and sharks (Figure 3), local synteny does not allow establishing the precise orthology relationships among lamprey and human genes (compare Figure 3 and Figure 6). Regarding the trees shown in Figure 4 and Figure 5, this alternative hypothesis would require postulating that *ASYN1* is a fast-evolving γ-syn gene. A high evolutionary rate would cause *ASYN1* to be highly divergent from the other agnathan γ-syn genes, which might lead to it appearing in an intermediate position in these trees, seemingly closer to *SNCA* and *SNCB* than to their true gnathostome ortholog *SNCG*. Such a proposal is equivalent to saying that the root of the tree would not be the one chosen to represent Figure 4 and Figure 5, but it would rather be located in the branch that separates gnathostome *SNCA* and *SNCB* genes from all the other sequences (indicated with an arrow in Figure 4 and Figure 5). It is relevant that statistical support for that alternative branch is non-significant in the tree shown in Figure 5 (bootstrap value is only 79%), while the most internal branch in Figure 4 and Figure 5, where the correct root would be if our original model is true, is strongly supported in both trees (99% and 97%, respectively). Nevertheless, it is clear that additional, independent data must be obtained to determine which of these two possible interpretations of the trees is correct.

Discriminating between these competing hypotheses is possible. Let us assume, based on our original hypothesis, that *ASYN1* is a true α/β-syn gene, located in *Petromyzon* on a chromosome (No. 41 in the current *P. marinus* assembly) that corresponds to the one that became duplicated in WGD2 to produce regions found today in human chromosomes 4 and 5. Also, let us hypothesize that *ASYN3* is the original γ-syn gene, located in *Petromyzon* in the chromosome (No. 8 in the *P. marinus* assembly) that became duplicated in WGD2 in gnathostomes, resulting in two chromosomes (Figure 1). One of these chromosomes is now part of chromosome 10 in our species, while the second one has been fragmented into smaller pieces that are currently found in different human chromosomes. If this is correct, genes in *Petromyzon* chromosome 41 should be most similar to those in human chromosomes 4 and 5, while genes on *Petromyzon* chromosome 8 should be most similar to those on human chromosome 10 and also to those found in the regions derived from the ancestral CLGQ/Pvc8 chromosome that were once found in the homolog of chromosome 10 produced in WGD2, but later became fragmented and distributed among several human chromosomes.

Characterizing the regions corresponding to the fragmented homolog of chromosome 10 has been attempted in several previous studies. A close relationship among many genes on human chromosomes 2, 4, 5, 8 and 10 was already detected by Dehal and Boore [83]. Subsequently, it was inferred that human chromosomes 2 and 8 contain fragments of the ancestral CLGQ/Pvc8 chromosome [62,63,84]. Obtaining evidence that human chromosomes 2 and 8 contain genes more closely related to those on chromosome 10 than to those on chromosomes 4 or 5 is quite straightforward. I downloaded the human data from the OHNOLOGS database [78] and determined the frequency of ohnologs located on specific pairs of human chromosomes (see Section 2). It turns out that the ohnologs characterized in that database are much more frequent in the pairs chromosome 2/chromosome 10 (23 cases) and chromosome 8/chromosome 10 (20 cases), compared to the pairs 2/4 (10 cases), 2/5 (4 cases), 8/4 (5 cases) and 8/5 (only 3 cases) (details in Appendix A). These results are in perfect agreement with the reconstruction of early events made by Simakov et al. and Nakatani et al. [62,63], which provided evidence that human chromosomes 2, 8 and 10 partly derive from a chromosome that resulted from the fusion of fragments of three ancestral vertebrate chromosomes (CLGQ/Pvc8, CLGF/Pvc7 and CGLI/Pvc9) that occurred after WGD1 but before WGD2. The existence of this mixed chromosome was independently deduced by Lamb [85]. On the other hand, the common regions in human chromosomes 4 and 5 would derive from a different ancient chromosome, in which regions of the ancestral CLGQ/Pvc8 and CLGI/Pvc9 chromosomes (but not CLGF/Pvc7) also fused after WGD1 but before WGD2 [62,63,85].

All these studies concluded that the regions derived from CLGQ/Pvc8 now found in human chromosomes 4 and 5, where *SNCA* and *SNCB* are located, started diverging just after WGD1 from the regions also derived from CLGQ/Pvc8 that are currently located in human chromosomes 2, 8 and 10 (being *SNCG* located on the latter chromosome). Keeping this in mind, we can now analyze the *Petromyzon* chromosomes, in order to discriminate between the two competing hypotheses proposed for the origin of the synuclein genes discussed earlier. If our original hypothesis is correct, *Petromyzon* chromosome 41, where the candidate α/β-syn gene *ASYN1* is located, should contain genes more similar to those on human chromosomes 4 and 5 than to those on human chromosomes 2, 8 and 10. On the other hand, *Petromyzon* chromosome 8, where the γ-syn gene *ASYN3* is present, should contain genes most similar to those on human chromosomes 2, 8 and 10, but less related to those on human chromosomes 4 and 5. However, if the alternative hypothesis is true, it means that all the genes in lampreys encode γ-synucleins. Then *Petromyzon* chromosomes 8 and 41 should both contain genes most similar to those in human chromosomes 2, 8 and 10. If such a result is indeed found, it would refute our original hypothesis.

Table 2 presents a summary of the main results obtained from comparing all the proteins encoded by genes on the relevant *Petromyzon* chromosomes with those in our species using BLASTP searches. For each protein, the three highest-scoring hits were recorded; the complete results can be found in Appendix A. In Table 2, three specific results are provided: the number of cases where the top score (column labeled “top”) or one of the three highest scores (“total” column) in the BLASTP searches using the *Petromyzon* sequences was found on each human chromosome and, finally, the number of cases where a single significant hit was obtained, i.e., when a single human gene showed similarity to the corresponding *Petromyzon* gene (“solo” column). Let us now examine the results for *Petromyzon* chromosomes 8 and 41. As expected, the most likely orthologs (“top” column”) of the genes found in those *Petromyzon* chromosomes are generally located on human chromosomes 2, 4, 5, 8, or 10. This is the case for 88% of the genes on *Petromyzon* chromosome 8 and 82% of the genes on *Petromyzon* chromosome 41. The exceptions can be attributed to either transposition of genes with an unrelated origin to the *Petromyzon* chromosomes, which occurred after the agnatha/gnathostoma split, or to gene losses in the human lineage, which eliminated the true orthologs of the *Petromyzon* genes. The results in the “top” and “total” columns indicate that *Petromyzon* chromosome 8 genes are statistically most similar (“top”) and most frequently related (“total”) to genes on human chromosome 10. On the other hand, genes on *Petromyzon* chromosome 41 are most often similar to those on human chromosome 4. The much lower values in both comparisons for chromosome 5, which in principle should be equivalent to chromosome 4, can be explained by a preferential retention of the genes on chromosome 4 over those on chromosome 5 after WGD2. Similarly, if we assume that WGD2 generated two identical chromosomes, one of them included now in human chromosome 10, while the other became fragmented and distributed across human chromosomes 2 and 8, it becomes evident that the genes on chromosome 10 have been preferentially retained, because adding up the values for the “top” and “total” columns for human chromosomes 2 and 8 gives values that are much smaller than that found for chromosome 10, although, interestingly, only slightly smaller than those for human chromosome 5 (Table 2).

Given the variation in the number of genes conserved since WGD1 in different chromosomes, the most effective way to test for differences between the lamprey chromosomes is by using 2 × 2 contingency tables. These tables consist of two rows obtained adding, respectively, the results for human chromosomes 4 and 5 and for chromosomes 2, 8 and 10, and two columns corresponding to each lamprey chromosome. Regarding *Petromyzon* chromosomes 8 and 41, there are significant differences between chromosomes for all three types of data (“top” data: chi-square value = 7.86, *p* = 0.005; “total” data: chi-square = 4.72, *p* = 0.030; “solo” data: chi-square = 4.94, *p* = 0.026; all chi-square tests have 1 degree of freedom). Consequently, the null hypothesis of equal similarity between the genes in lamprey chromosomes 8 and 41 and the human genes that come from the two homologous chromosomes derived from CLGQ/Pvc8, which would support the idea that *ASYN1* and *ASYN3* are both γ-synuclein-coding genes, is rejected. These results, which demonstrate a significant excess of related genes in human chromosomes 4 and 5 with those on lamprey chromosome 41, are fully consistent with our original hypothesis, which implied that *ASYN1* is an α/β-syn gene, but not with the alternative hypothesis postulating *ASYN1* as a γ-syn gene.

The “solo” column in Table 2 is particularly significant. While the “top” and “total” results indicate general trends regarding the similarities among the human and lamprey genes, the “solo” column accounts for gene losses, which are rare, unique events. There are several scenarios in which genes detected in a specific *Petromyzon* chromosome may be found isolated, without any duplicates, in humans (Figure 7; see next page). Alternatives (a) and (b) in Figure 7 involve two independent gene losses, while three losses are required in Figure 7c. It is noteworthy that, for a given *Petromyzon* gene, the probability of the single similar gene found in humans being either an ortholog or a paralog is the same for alternatives (b) or (c). On the contrary, for alternative (a), in which the loss occurs after WGD1 in the ancestor of all vertebrates, the gene found in humans must necessarily be an ortholog of the lamprey gene. This asymmetry is very significant in our context, because it indicates a systematic bias towards “solo” genes being true orthologs, which can therefore be used as markers to establish precise correspondences between lamprey and human chromosomes. Therefore, finding a significant deviation in the frequencies of “solo” genes from what is expected by chance (as demonstrated earlier) provides strong evidence that *Petromyzon* chromosome 8 corresponds to the regions now located on human chromosomes 2, 8 and 10, and *Petromyzon* chromosome 41 corresponding to regions of human chromosomes 4 and 5. An examination of Appendix A allows establishing in more detail the exact relationships between the *Petromyzon* and human chromosomes. The similarities with the lamprey genes are largely restricted to human genes located on regions 2p11-13, 4q13-25, 5q31-35, 8p11-22 and 10q11-26. This indicates that the CLGQ/Pvc8-derived genes that after WGD1 became located on chromosomes 8 and 41 of *Petromyzon* are now found in humans throughout the large arms of chromosomes 4 and 10, as well as in relatively small regions of the large arm of chromosome 5 and the short arms of chromosome 2 and 8.

### 3.3. Prediction 3: Additional Synuclein Genes Are Likely to Be Found in Agnathan Vertebrates, Given Their Lineage-Specific Hexaploidization

We may now explore the origin of the other two *ASYN* genes, which appear in the trees as close relatives to the γ-syn gene *ASYN3* (Figure 4 and Figure 5). For *ASYN4*, which is by far the most divergent, the lack of genomic data precludes further investigating its origin. However, *ASYN2* is located on *Petromyzon* chromosome 22, for which data are available that allow the same comparisons with human chromosomes already performed for chromosomes 8 and 41. These comparisons are also summarized in Table 2. It turns out that most genes in that chromosome have the highest similarity (“top” column) to genes
found on human chromosomes 1 and 9. Only about 18% show top scores with genes on human chromosomes 2, 4, 5, 8, or 10. This indicates that *Petromyzon* chromosome 22 has a complex origin, with only a small fragment originating from the ancestral CLGQ/Pvc8 chromosome. This fact was already discovered by Simakov et al. and Nakatani et al. [62,63]. Given the strong similarity between *ASYN2* and *ASYN3*, a possible explanation is that the CLGQ/Pvc8 fragment of lamprey chromosome 22 is derived from a duplication of part of *Petromyzon* chromosome 8, perhaps as a consequence of the additional, agnathan-specific WGDs. If this is true, the *Petromyzon* chromosome 22 genes derived from the CLGQ/Pvc8 ancestral chromosome should most often show similarity to genes located on human chromosome 10 and the pattern of “top”, “total” and “solo” hits should be identical to that found for lamprey chromosome 8. This is indeed what we observe (Table 2). The analyses of 2 × 2 contingency tables comparing chromosome 22 with either chromosome 8 or chromosome 41 showed that, as expected, the null hypothesis is rejected for all comparisons involving lamprey chromosomes 22 and 41, while it is accepted, also as expected if the regions have a related origin, for the “top” (chi-square = 2.31, *p* = 0.128) and “solo” (chi-square = 2.58, *p* = 0.108) comparisons for lamprey chromosomes 8 and 22. However, the null hypothesis is rejected for the “total” comparison of chromosomes 8 and 22 (chi-square = 5.43, *p* = 0.020), due to an excess of positive results in the comparisons of lamprey chromosome 22 and human chromosomes 2 and 8 (Table 2). This anomaly is easily explained by the mixed origin of these three chromosomes. In the large regions not derived from CLGQ/Pvc8 that they contain, some genes with weak similarity, e.g., distantly related paralogs, certainly must exist. This would not affect the “top” or “solo” data, where only strong or unique similarities, almost certainly due to a common CLGQ/Pvc8 origin, are counted, but it will artificially increase the similarity among these three chromosomes shown in the “total” column. In summary, both sequence similarity (Figure 4 and Figure 5) and global synteny data (Table 2) indicate that *ASYN2* is an agnathan-specific *SNCG* duplicate. The fact that *Petromyzon* chromosome 22 contains 37 genes derived from the ancestral chromosome CLGQ/Pvc8 (Table 2) strongly argues against *ASYN2* being the product of an individual gene duplication. In fact, when searching for paralogs of those 37 genes in *P. marinus*, it was found that, out of 31 genes for which paralogs were detected, in sixteen cases the closest paralog was either on chromosomes 8 (11 cases, including the *ASYN2/ASYN3* pair) or chromosome 41 (5 cases) (Appendix A). This is the expected result if the CLGQ/Pvc8-derived regions on chromosome 22 originated from either a large segmental duplication or a whole-genome duplication. Given our current knowledge, the simplest explanation is that *ASYN2* emerged in the agnathan-specific genome hexaploidization.

We may now summarize the results obtained so far, to show that they all are in good agreement with the predictions of the model depicted in Figure 1. First, the gnathostome ancestor already possessed *SNCA*, *SNCB* and *SNCG* genes. Almost all gnathostomes have these three genes, with the exceptions resulting from secondary, lineage-specific losses or duplications (Prediction 1). Second, lampreys not only have genes encoding proteins with strong similarity with gnathostome γ-synucleins but also a gene, *ASYN1*, which encodes a protein that is most similar to both α- and β-synucleins (Prediction 2). Third, these lamprey genes are located on chromosomes that contain regions that derived from a single ancestral chromosome, CLGQ/Pvc8, and the sets of genes present in each agnathan chromosome are related to those in gnathostomes exactly as predicted in our model (also in agreement with Prediction 2). Finally, two additional synuclein genes have been detected in lampreys, and the available evidence indicates that at least one of them originated in the agnathan-specific hexaploidization (Prediction 3).

### 3.4. Structural Considerations

Characterizing highly divergent *SNC* genes provides intriguing new insights into the changes that synuclein proteins can undergo. The N-terminal repeat regions of the osteichthyan proteins analyzed prior to this study were quite homogeneous (see, e.g., [1,2,3,6,25,27]). However, when examining both chondrichthyan and agnathan proteins, a much greater diversity is observed (Figure 8; Appendix A). The first interesting feature is the variability in the center of that region. In both the shark *SNCB*- and lamprey *ASYN1*-derived proteins, some amino acids are missing (Figure 8). Also, at least five independently generated small deletions can be found in that region in the 26 osteichthyan α-syn and β-syn proteins analyzed (Appendix A). Given that this region is much more variable in length than previously assumed, it is reasonable to postulate that it initially consisted of typical 11-amino-acid-long repeats but is highly prone to suffer partial deletions. This may be due to the fact that it is precisely in this region (specifically, amino acids corresponding to repeat 4 in Figure 8), where the helical structure of the N-terminus of synucleins breaks under some physiological conditions, generating two small α-helices instead of a single, continuous one [14,15]. If this hypothesis is true, it would mean that the N-terminal region of synucleins originally did not have seven repeats, as generally assumed, but eight (as shown in Figure 8, bottom). The current pattern would have evolved in two steps: (1) a seven-amino-acid-long deletion, eliminating part of repeats 4 and 5, which happened before WGD1 (and it is thus found in all synucleins); and (2) additional deletions leading to a further shortening of these repeats in many lineages. Notably, in multiple cases (*ASYN1* in lampreys, *SNCA* in *Polypterus*, *Takifugu* and *Acipenser*; *SNCB* in *Takifugu* and *Danio*), these recent deletions span four amino acids, leading to proteins that have lost exactly 11 amino acids, i.e., a complete repeat (see *Petromyzon* *ASYN1* in Figure 8; the rest in Appendix A).

A second discovery may also have significant functional implications. The only differential feature described so far in the repeats region was that all analyzed β-synucleins had, when compared with the other synuclein proteins, a specific, 11-amino-acid-long deletion, again equivalent to eliminating one of the repeats. This deletion has been suggested to be important for reducing the likelihood of β-synuclein proteins aggregating relative to α-synucleins, as it shortens the NAC region [50]. However, surprisingly, chondrichthyan β-synucleins do not have that deletion (Figure 8; Appendix A). Thus, a shortened NAC peptide is not essential, at least in those species, for proper β-synuclein function. Finally, a third interesting structural result is that the highly divergent proteins encoded by the lamprey gene *ASYN4* apparently lack the typical C-terminal region that exists in all other synucleins (Figure 8, Appendix A).

## 4. Discussion

The recent characterization of several agnathan genomes, together with the precise knowledge of when the two vertebrate WGDs occurred, allows for the determination of the ohnologs present in all vertebrate lineages. In this study, I have combined information from sequence comparisons and gene locations to establish that the vertebrate genome duplications played a critical role in the emergence of the synuclein genes. The methods used here, involving a combination of phylogenetic trees, synteny analysis and whole-chromosome statistical comparisons to confirm orthology relationships, can be considered a model to follow in other studies of challenging gene families, whenever sequence analyses alone are deemed insufficient. Regarding the phylogenetic analyses, one of the main strengths of this work is the selection of a set of model species that covers the entire vertebrate spectrum. In particular, the inclusion of several types of osteichthyan fishes so far neglected, such as bichirs, sturgeons and lungfishes, as well as a careful representation of chondrichthyans and agnathans, have been crucial in establishing clear conclusions about the early evolution of synucleins. A second strength is the comparison of the results provided by different sequence alignment algorithms. This is particularly important given the limited information provided by the synuclein protein sequences. Here, the final outcome of the ML analyses has been used to select the most adequate algorithm among 21 alternatives. Notice that alignments and tree reconstructions have been considered together as a single unit, a unified model to explain how these sequences evolved. The comparison of the phylogenetic trees with all the other independent information obtained, synteny data and similarity among chromosomes, has confirmed that those trees make perfect sense, thus validating our analytical strategy. The only unexpected results concerned the ambiguous position of the β-syn chondrichthyan sequences when analyzing truncated versions of the synuclein proteins (Figure 2a and Figure 4). This ambiguity arises from the combination of two factors. First, the truncated sequences are very short (less than 100 amino acids). Second, the sequences have a convergent feature that distorts the analysis: chondrichthyan β-syn sequences include eleven amino acids that are absent in all other vertebrate β-synucleins but present in α-synucleins (Figure 8; Appendix A). However, when the whole sequences are considered (see Figure 2b and Figure 5), the chondrichthyan β-syn sequences appear exactly in the expected positions. The relatively long C-terminal tails of all β-synucleins are very similar (Appendix A), which compensates for the statistical anomaly caused by the 11-amino-acid-long deletion.

The results provided by the trees for the agnathan sequences (Figure 4 and Figure 5) are consistent with those species having both α/β- and γ-synucleins, as expected according to our hypothesis that all vertebrates synucleins are encoded by ohnologs (Figure 1). However, formally demonstrating this has been complicated because, unlike in gnathostomes (Figure 3), it is not possible to use local synteny to confirm the orthologies suggested by the phylogenetic trees (as can be deduced by comparing Figure 3 and Figure 6; see Section 3). For that reason, a different strategy has been employed. Given the data indicating that fragments of human chromosomes 4, 5 and 10, where our synuclein genes are located, as well as parts of *Petromyzon marinus* chromosomes 8, 22 and 41, where tree of the four lamprey synuclein genes are found, all originate from the same ancestral vertebrate chromosome [62,63], global comparisons of those chromosomes have been conducted to infer the exact orthology relationships for agnathan and gnathostome synucleins. The comparisons of all proteins in these chromosomes have demonstrated that *Petromyzon marinus* chromosome 41, where its potential α/β-syn gene *ASYN1* is found, contains many genes that are most similar to those on human chromosomes 4 or 5, where our α- and β-synuclein genes are located (Table 2). On the other hand, chromosomes 8 and 22 of *P. marinus* contain genes that are, in statistical terms, more similar to those on human chromosomes 2, 8 and 10 than to those on chromosomes 4 and 5 (see also Table 2). This is the expected result if the *Petromyzon* genes *ASYN2* and *ASYN3* are both orthologs of the gnathostome *SNCG* gene, which encodes γ-synuclein. These results, along with the phylogenetic analyses, refute the hypothesis that vertebrate synucleins were produced by conventional duplications from an ancestral γ-synuclein gene [7,58] as well as another potential explanation, formulated here for the first time, which implied that three of the agnathan synuclein genes derived from agnathan-specific genome duplications. The only explanation that fits well with all the available data is that all gnathostome SNC genes and at least two of the four synuclein-coding agnathan genes (*ASYN1* and *ASYN3*) originated from the classical vertebrate WGDs. Evidence for lamprey *ASYN2* suggests that it may be the product of another WGD that occurred after the agnathan/gnathostome split [63]. Data for the fourth lamprey gene, *ASYN4*, are still insufficient to conclude its origin, but its close sequence similarity to *ASYN3* (Figure 4 and Figure 5) suggests that it is another γ-synuclein duplicate, perhaps also emerged in the agnathan-specific WGDs. These results can be used to reconsider previous studies using agnathan species. Busch and Morgan [59] characterized three *Petromyzon marinus* synuclein genes, corresponding to *ASYN1* (which they called “Lamprey Syn 3”), *ASYN2* (“lamprey γ-syn (FD)”) and *ASYN3* (“lamprey γ-syn (DY)”). They correctly identified the last two as coding for gamma synucleins, but could not assign a gnathostome ortholog for *ASYN1*. They found that *ASYN3* is the most abundantly expressed in lamprey brain, and its expression increased after spinal cord injury in some particular neurons, which often later died [59]. Down-regulation of *ASYN3* increased neuron survival [86]. On the other hand, Vorotnstova et al. [60] characterized the same three synuclein genes in another lamprey species, *Lampetra fluviatilis*, although they interpreted all three as coding for gamma synucleins. The fourth agnathan gene, *ASYN4*, was not detected in those earlier studies. A detailed analysis of the expression patterns of the lamprey genes has yet to be published. Determining the functions of *ASYN1* may be particularly interesting, as it may shed light on the origin of the current functions of gnathostome α- and β-synucleins.

The precise characterization of when each synuclein emerged allows us to contextualize several significant evolutionary events. For example, we can now establish the exact timing of the different deletions that affect the center of the repetitious N-terminal region or the deletion that occurred in osteichthyan β-synucleins (Figure 8; Appendix A). Another interesting evolutionary deduction concerns the genes that surrounded the first ancestral synuclein gene. The comparison of lamprey, shark and human sequences allows postulating a cluster of seven adjacent genes, including that original *SNC* gene, before WGD1 (see above). This knowledge may be used to search for functional relationships among these genes and perhaps to determine the mysterious origin of that ancestral synuclein gene, if it is somehow derived from one of these adjacent genes. Finally, the fact that three *SNCA* gene losses have been detected in the model gnathostome species studied here, but no *SNCB* or *SNCG* deletions have been found, is intriguing. In principle, there is no reason why *SNCB* or *SNCG* losses of function should be less common. The three synuclein-coding genes have been independently inactivated in mice, resulting in subtle changes in otherwise viable and fertile individuals, who do not exhibit major anomalies in their nervous systems [34,87,88,89,90].

Another interesting implication of this work is that most, if not all, synuclein genes did not experience any dosage effects when they first appeared. The three canonical gnathostome *SNC* genes, three or perhaps all four duplicates detected in agnathans, plus the *Takifugu/Danio* and *Acipenser* duplicated *SNCG* genes, can all be linked to WGDs, which do not involve dosage changes among functional partners. Since both synuclein gene duplications and single, dominant amino acid changes in synuclein proteins can, in some cases, lead to protein aggregation, cell damage and disease, WGDs may be by far the simplest way for synuclein diversification [91,92,93,94,95]. These considerations naturally raise the question of whether other Parkinson’s disease-related genes also emerged in the vertebrate WGDs, in parallel with synucleins, perhaps also to avoid dosage effects. Significantly, this has been proven false for all the genes that have been thoroughly studied, namely *PRKN* [96,97], *LRRK2* [98,99], *DJ-1* [100], *PINK1* [101] and *ATP13A2* [102], which are all ancient genes, emerged before WGD1. The rest have not been properly analyzed, but it is possible that some of them are indeed the product of the vertebrate WGDs. In the Ohnologs database, several candidates (e.g., *UCHL1*, *GIGYF2*) can be found, although specific studies would be required to confirm their status as true ohnologs. These studies may be relevant, because obtaining a more comprehensive view of when the Parkinson’s disease-related genes emerged could contribute to our understanding of their functional relationships.

## 5. Conclusions

The origin of the synuclein genes present in both agnathans and gnathostomes can be explained by a single model (Figure 1). Their emergence is associated with the whole-genome duplications that occurred throughout vertebrate history. This may be attributed to synuclein proteins being prone to cause cellular damage if their dosage is altered. Furthermore, the structural variability of the synuclein proteins is greater than previously assumed.

## Figures and Tables

**Figure 1 biology-12-01053-f001:**
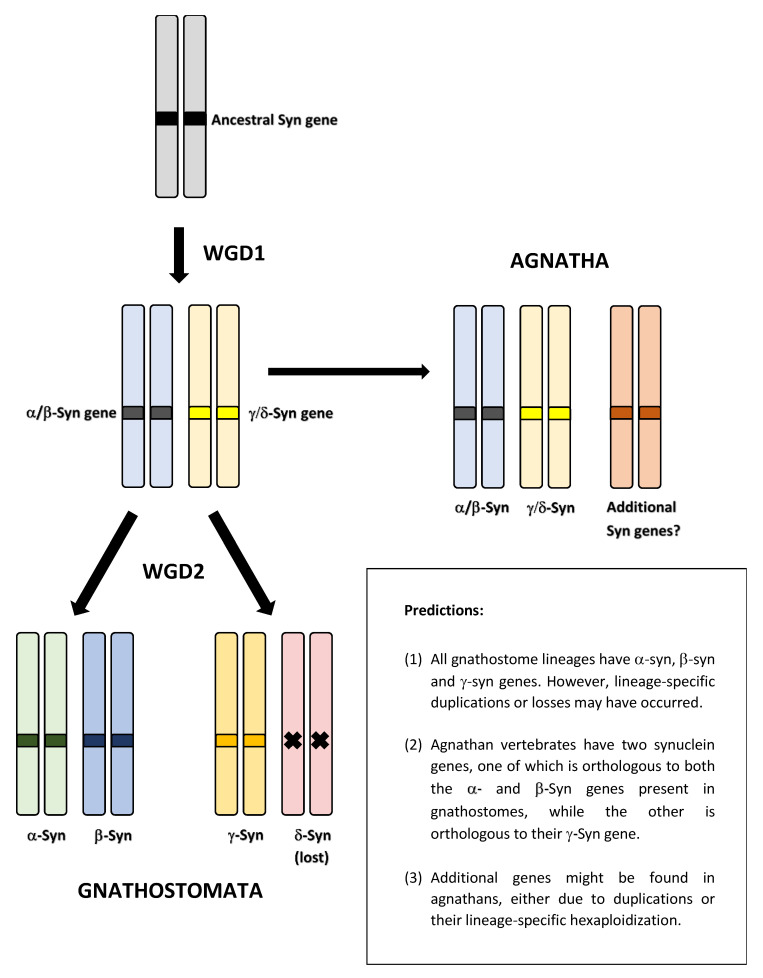
Model of synuclein emergence linked to the vertebrate WGDs and predictions derived from that model. A single synuclein-coding gene was present in the ancestor of all vertebrates. WGD1 generated two different genes prior to the agnathan/gnathostome split. In the gnathostome lineage, WGD2 generated four synuclein genes, one of which (“δ-syn”) was lost before gnathostome diversification. Agnathans may have additional duplicates due to the hexaploidization of their genomes.

**Figure 2 biology-12-01053-f002:**
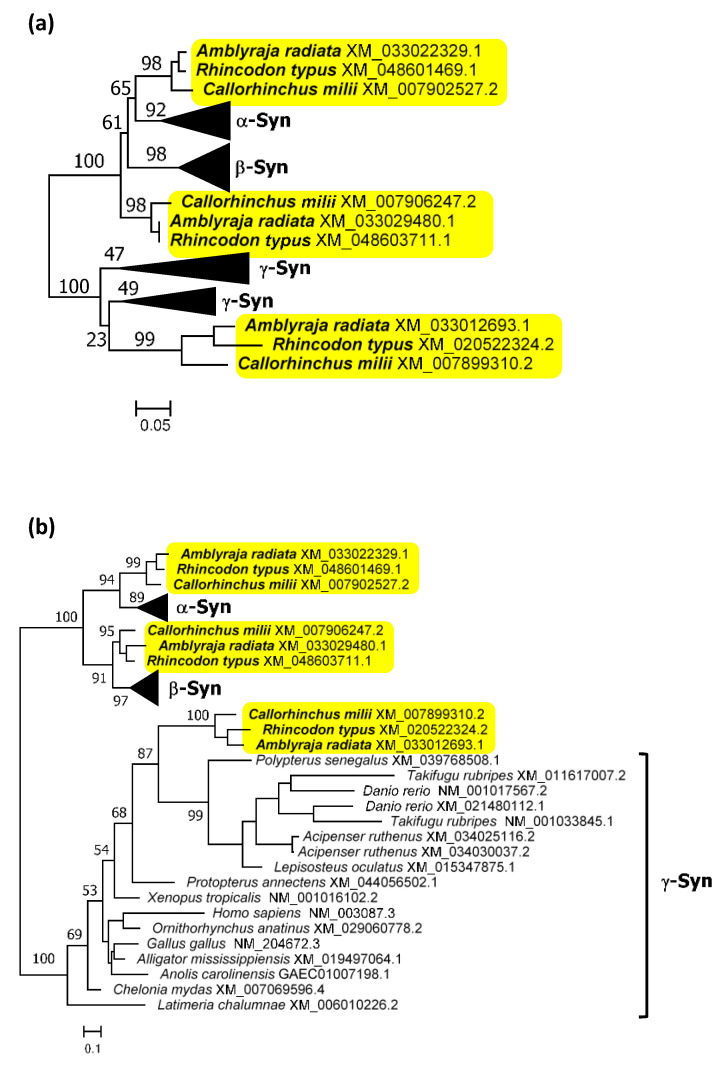
Maximum-likelihood trees obtained from the analysis of gnathostome synuclein proteins. Species names and accession numbers are used for sequence identification; the complete list is available in Appendix A. Numbers indicate the percentage of bootstrap support for the respective branches. For simplification, bootstrap values are indicated only for the most external branches, critical for interpreting the trees. In yellow, chondrichthyan sequences. When possible, the osteichthyan sequences have been collapsed into groups. (**a**) Optimal tree obtained from an alignment of the conserved N-terminal region of synucleins (alignment algorithm: MAFFT FF-TNS-2; model of sequence evolution: Q.plant+G4; *pers* parameter = 0.2; LnL = −2287.782). Four groups of gnathostome sequences are shown. (**b**) Tree generated using the entire sequence (alignment algorithm: MAFFT FF-TNS-2; Model of sequence evolution: Q.plant+G4; *pers* = 0.5; LnL = −4914.911). Gnathostome α-syn and β-syn sequences have been grouped together. The topology of the tree precludes doing the same for their γ-syn sequences.

**Figure 3 biology-12-01053-f003:**
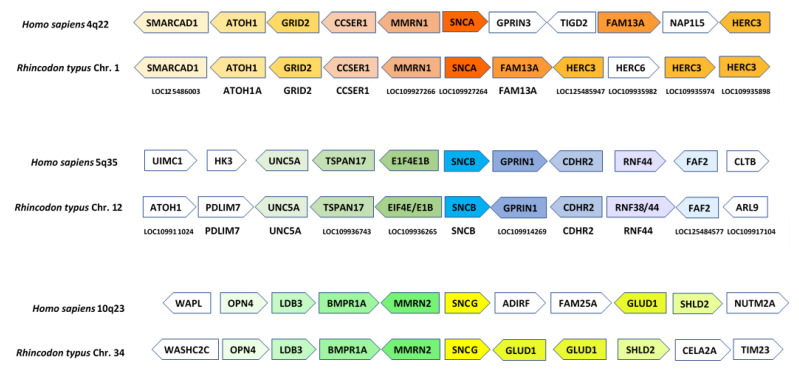
Synteny analyses comparing the regions around the three synuclein-coding genes in humans and the shark *Rhincodon typus*. In each region, the arrows indicate either the human genes (top line of arrows) or the most similar human genes when the *R. typus* genes are compared (BLASTP searches) against the complete set of human proteins (bottom line). The names of the *R. typus* genes are indicated below both lines of arrows. The orientations of the arrows reflect those of the genes on their respective chromosomes. Colors have been added to highlight potential orthologies.

**Figure 4 biology-12-01053-f004:**
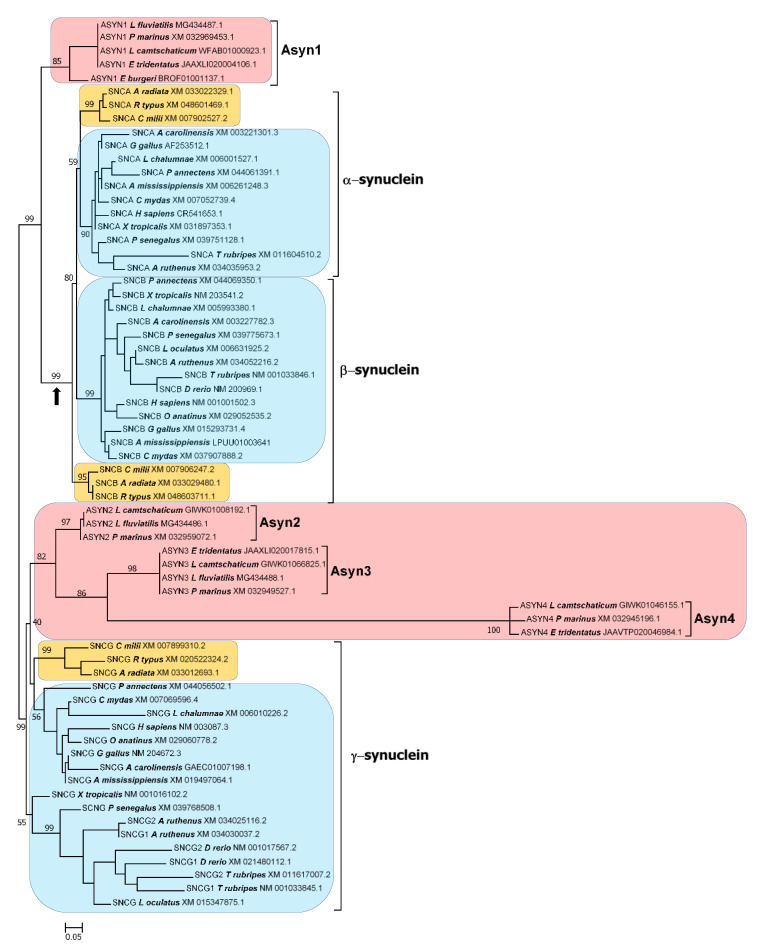
Maximum-likelihood tree obtained by comparing the N-terminal regions of synucleins from the 22 model species. Agnathan, chondrichthyan and osteichthyan proteins are indicated in magenta, orange and blue, respectively. Bootstrap values are indicated as in Figure 2; again, only the values for the critical branches are detailed. An arrow indicates an alternative root of the tree (see main text). Alignment algorithm used in the tree: MAFFT FF-TNS-2; model of sequence evolution: Q.plant+G4; *pers* parameter = 0.2; LnL = −2845.072.

**Figure 5 biology-12-01053-f005:**
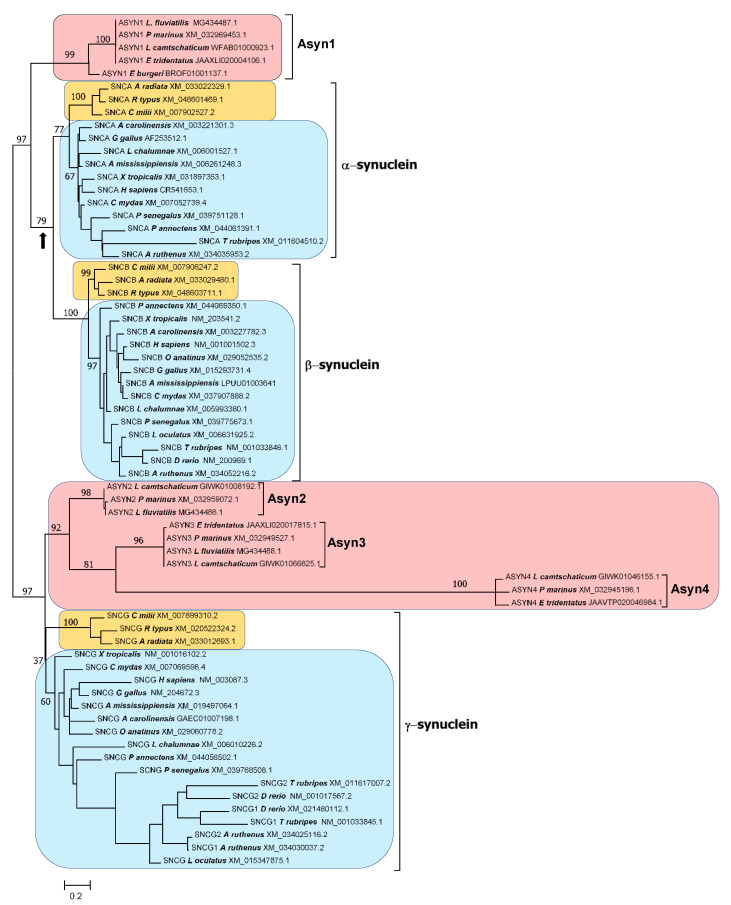
Maximum-likelihood tree obtained for the synucleins of the twenty-two model species, when the full sequences are analyzed. Conventions are the same as in Figure 4. The arrow once again indicates the alternative root of the tree discussed in the main text. Alignment algorithm: MAFFT FF-TNS-2; model of sequence evolution: Q.plant+G4; *pers* parameter = 0.5; LnL = −5816.227.

**Figure 6 biology-12-01053-f006:**
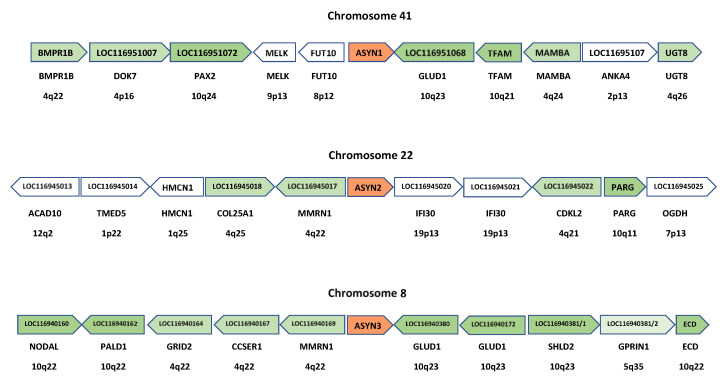
Genes surrounding the synuclein-coding ones in *Petromyzon marinus*. The names within the arrows indicate the names of the lamprey genes. Below the arrows, both the names of the most likely human orthologs and their chromosomal positions are provided. These are the ones with the highest scores when the lamprey genes were compared (BLASTP) with the complete set of human proteins. Lamprey *ASYN1-ASYN3* genes are indicated in red. Lamprey genes with potential orthologs on human chromosomes 4, 5 or 10 are highlighted in green.

**Figure 7 biology-12-01053-f007:**
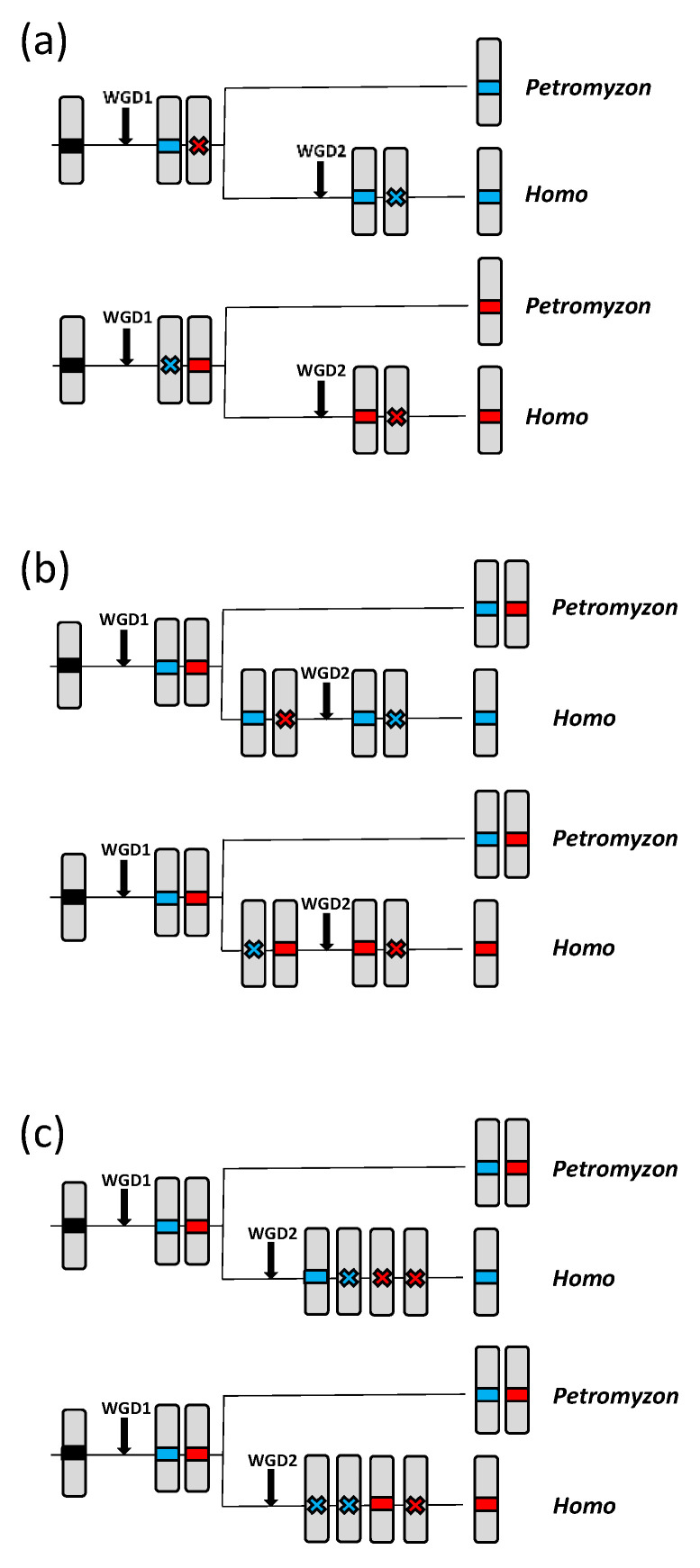
Models explaining the events leading to a single human gene showing significant similarity when compared with a given agnathan gene. Red and blue boxes are used to identify the chromosomes; only one of the two homologous chromosomes is depicted. (**a**) A gene loss occurred after WGD1 but before the agnathan/gnathostome split, followed by another loss after WGD2 in gnathostomes. In both cases, the lamprey and human genes are necessarily orthologs. (**b**) Two losses occurred in the gnathostome lineage, one before and the second after WGD2. In this scenario, there is a 50% probability of the human gene being an ortholog and 50% of being a paralog of a given agnathan gene. (**c**) Three losses occurred in gnathostomes after WGD2. Again, there is a 50% probability of the remaining human gene being either an ortholog or a paralog of a given lamprey gene. This evolutionary history is expected to be less frequent than those in (**a**) or (**b**), which require just two gene losses.

**Figure 8 biology-12-01053-f008:**
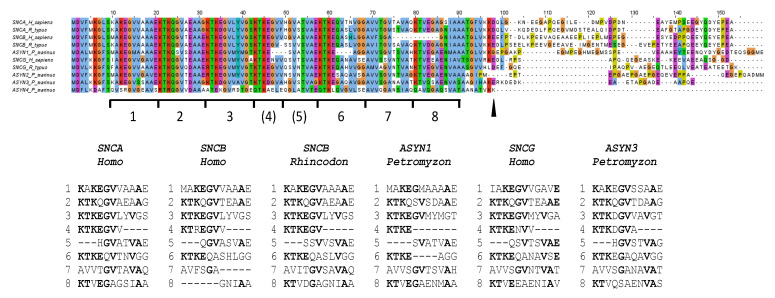
Alignment of selected sequences and schematic representation of the eight repeats hypothesized in synuclein sequences. Repeats 4 and 5 would have been truncated by a 7-amino-acid-long deletion that occurred very early in synuclein evolution. The arrowhead indicates the end of the highly conserved region present in all synucleins. The trees shown in Figure 2a and Figure 4 were obtained with sequences truncated at that point.

**Table 1 biology-12-01053-t001:** Model species analyzed in this study. Species names (in bold, italics) and taxa to which they belong are indicated. Magenta: agnathan species. Yellow: chondrichthyan species. Blue: osteichthyan species.

** *Petromyzon marinus* **	Cyclostomata	Hyperoartia
** *Lethenteron camtschaticum* **	Cyclostomata	Hyperoartia
** *Lampetra fluviatilis* **	Cyclostomata	Hyperoartia
** *Entosphenus tridentatus* **	Cyclostomata	Hyperoartia
** *Eptatretus burgeri* **	Cyclostomata	Myxini
** *Callorhinchus milii* **	Gnathostomata	Chondrichthyes; Holocephali
** *Amblyraja radiata* **	Gnathostomata	Chondrichthyes; Elasmobranchii; Batoidea
** *Rhincodon typus* **	Gnathostomata	Chondrichthyes; Elasmobranchii; Selachi
** *Polypterus senegalus* **	Gnathostomata	Actinopterygii; Cladistia
** *Acipenser ruthenus* **	Gnathostomata	Actinopterygii; Actinopteri; Chondrostei
** *Lepisosteus oculatus* **	Gnathostomata	Actinopterygii; Actinopteri; Neopterygii; Holostei
** *Takifugu rubripes* **	Gnathostomata	Actinopterygii; Actinopteri; Neopterygii; Teleostei
** *Danio rerio* **	Gnathostomata	Actinopterygii; Actinopteri; Neopterygii; Teleostei
** *Latimeria chalumnae* **	Gnathostomata	Sarcopterygii; Coelacanthimorpha
** *Protopterus annectens* **	Gnathostomata	Sarcopterygii; Dipnomorpha
** *Xenopus tropicalis* **	Gnathostomata	Sarcopterygii; Amphibia
** *Anolis carolinensis* **	Gnathostomata	Sarcopterygii; Amniota; Lepidosauria
** *Chelonia mydas* **	Gnathostomata	Sarcopterygii; Amniota; Archelosauria; Testudinata
** *Alligator mississippiensis* **	Gnathostomata	Sarcopterygii; Amniota; Archelosauria; Archosauria; Crocodylia
** *Gallus gallus* **	Gnathostomata	Sarcopterygii; Amniota; Archelosauria; Archosauria; Dinosauria
** *Ornythorhynchus anatinus* **	Gnathostomata	Sarcopterygii; Amniota; Mammalia; Prototheria
** *Homo sapiens* **	Gnathostomata	Sarcopterygii; Amniota; Mammalia; Theria

**Table 2 biology-12-01053-t002:** Comparisons of genes located on chromosomes 8, 22 and 41 of *Petromyzon marinus* with the human genes. Colors are used to distinguish, according to their origin, the human chromosomes relevant to this study.

	*Petromyzon* chr. 8			*Petromyzon* chr. 41			*Petromyzon* chr. 22		
Human Chromosome	Top	Total	Solo	Top	Total	Solo	Top	Total	Solo
1	2	12	0	5	13	2	68	109	19
2	8	29	1	7	26	1	2	14	1
3	0	12	0	1	8	0	0	8	0
4	59	96	4	61	92	10	8	16	0
5	14	48	1	20	48	5	4	14	0
6	3	10	0	1	3	0	11	25	3
7	2	11	0	2	6	2	3	11	0
8	2	18	1	5	12	0	6	15	2
9	4	10	2	4	12	0	61	98	14
10	75	117	12	36	72	9	17	32	5
11	0	9	0	1	9	0	1	5	0
12	1	8	0	2	7	0	2	13	0
13	0	2	0	2	4	0	1	2	0
14	0	3	0	1	4	0	0	6	0
15	0	1	0	1	5	0	1	6	1
16	0	8	0	0	5	0	2	10	1
17	0	6	0	3	8	2	1	13	1
18	0	0	0	0	0	0	2	4	0
19	1	7	0	2	13	0	12	37	4
20	6	10	2	3	6	1	1	5	1
21	1	4	0	0	2	0	0	1	0
22	0	2	0	0	1	0	0	2	0
X	2	8	0	0	2	0	1	5	0

## Data Availability

All data required to repeat the analyses shown in this study are available either in the Appendix A or in the public databases cited in the text.

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
