# Peer review of "Emergence of the Synucleins"

_biology, 2023, doi:10.3390/biology12081053_

Round 1

Reviewer 1 Report

The manuscript authored by Ignacio Marin delves into the investigation of the origin and evolution of synuclein genes in all vertebrates through the utilization of phylogenetic analysis and synteny. In this study, the author presents intriguing findings that shed light on the origin of synuclein genes found in both agnathans and gnathostomes, using a single model system. Moreover, the author puts forth the proposition that the diversification of synuclein genes in all living vertebrates can be attributed to whole-genome duplications.

While the manuscript offers valuable insights, there are a couple of areas where further attention would enhance its clarity and comprehensibility. Firstly, it would be beneficial if the author provided a detailed analysis of the structural divergence of SNC proteins between the selected species. This additional information would contribute to a deeper understanding of the variations and potential functional implications of synuclein proteins across different vertebrates.

Furthermore, in Figure 8 of the manuscript, it is recommended that the author incorporate a multiple sequence alignment in color, with the sequences being highlighted in lighter shades. This adjustment would greatly enhance the visual clarity of the figure, allowing readers to more easily discern the distinctions and similarities between the aligned sequences.

By addressing these suggestions, the manuscript would be further strengthened, ultimately providing a more comprehensive and insightful exploration of the origin and evolution of synuclein genes in vertebrates

Reviewer 2 Report

This article entitled “Emergence of the synucleins” by Ignacio Marín provides a phylogenetic analysis of the vertebrate synucleins using a well sampled dataset comprised of vertebrate genomes. Importantly the article addresses several hypotheses on the origins of synucleins related to whole genome duplications that have occurred in the evolutionary history of vertebrates by way of synteny analysis. The paper is well written and more comprehensive than previous studies. In addition, while I am not an expert in synucleins, the paper seems to be well researched and appropriate citations are given. I have no major concerns with the manuscript; however, I outline a couple of minor concerns below that could be addressed to improve and add clarity to the manuscript.

Minor Comments:

Line 250. Phylogenetic analyses of molecular sequences provide hypotheses on relationships, but do not prove such relationships. Here, analyses offer improved support for specific phylogenetic hypotheses, but do not prove them even if nodal support is maximum.

Fig 2. It is not clear why there are different numbers of operational taxonomic units in 2a and 2b. Perhaps it is because additional clades are collapsed in 2a? This could be explained better.

Line 316. “ASYN1 is most similar to the SNCA and SNCB gnathostome genes …”. Here you are detecting orthology between ASYN and the clade containing SNCA and SNCB. The term “similar” is erroneous, you are detecting orthology based on a topology determined by a probabilistic model for molecular evolution.

Reviewer 3 Report

In the article titled "Emergence of the synucleins," the author examines the origin of different synuclein genes present in the human genome. This investigation is conducted through an assessment of 22 model species and whole-genome duplication events. The author's conclusion is that two specific whole-genome duplications led to the emergence of the three synuclein genes found in humans.

While the article's theme is well formulated, the formulated predictions can be challenging to follow. Nevertheless, the manuscript is comprehensive, providing a reasonable amount of detail on the problem, hypothesis, methodology, and results. We have a few specific inquiries for the author:

  1. It would be helpful to include a summary of the taxa tree, including the events of WGD1 and WGD2, to aid readers in following the author's hypothesis. For example, it is difficult for a researcher interested in SNCA to determine which species are chondrichthyans or osteichthyans in Figure 2. This information is easier to comprehend in Figures 4 and 5.

  2. The mention of jawless vertebrates uniquely in the prediction title adds unnecessary complexity to the manuscript's readability.

  3. In Page 5, lines 165-167, the author mentions that protein sequences were retrieved from nr, wgs, est, and tsa databases. We would like to know if any divergent information was observed, and how the author decided on the final sequence(s) to use.

  4. In Page 5, lines 175-178, could the author provide more information about the default mapping parameters for ClustalX? As these parameters are likely to change in the future, reproducing the analysis may become complex.

  5. In Page 5, lines 206-208, could the author explain why the three species were chosen for the study?

  6. In Page 9, lines 320-332, the logic becomes difficult to follow as the species mentioned are not clearly defined as Gnathostomata or Agnatha.

  7. In Page 19, line 651, the author mentions studies suggesting that the N-terminal repeat regions of osteichthyan proteins were homogeneous. We would like to know which studies propose this and what could explain the divergent analysis.

Round 2

Reviewer 1 Report

All questions were addressed.